**www.cambridge.org/ext**

## Research Article

Permian; Triassic; mass extinction; climate crisis; thermal stress; invertebrates

**Corresponding author:**
William J. Foster;
Email: w.j.foster@gmx.co.uk

# Thermal and nutrient stress drove Permian–Triassic shallow marine extinctions

William J. Foster[1] , Anja B. Frank[1] , Qijian Li[2] , Silvia Danise[3], Xia Wang[4] and Jörn Peckmann[1]

[1]Universität Hamburg, Institut für Geologie, Hamburg, Germany; [2]State Key Laboratory of Palaeobiology and Stratigraphy, Nanjing Institute of Geology and Palaeontology, CAS, Nanjing, Jiangsu, China; [3]Dipartimento di Scienze della Terra, Università degli Studi di Firenze, Via La Pira 4, 50121 Firenze, Italy and [4]Institute of Sedimentary Geology, Chengdu University of Technology, Chengdu, Sichuan, China

## Abstract

The Permian–Triassic climate crisis can provide key insights into the potential impact of horizon threats to modern-day biodiversity. This crisis coincides with the same extensive environmental changes that threaten modern marine ecosystems (i.e., thermal stress, deoxygenation and ocean acidification), but the primary drivers of extinction are currently unknown. To understand which factors caused extinctions, we conducted a data analysis to quantify the relationship (anomalies, state-shifts and trends) between geochemical proxies and the fossil record at the most intensively studied locality for this event, the Meishan section, China. We found that $\delta^{18}O_{apatite}$ (paleotemperature proxy) and $\delta^{114/110}Cd$ (primary productivity proxy) best explain changes in species diversity and species composition in Meishan's paleoequatorial setting. These findings suggest that the physiological stresses induced by ocean warming and nutrient availability played a predominant role in driving equatorial marine extinctions during the Permian–Triassic event. This research enhances our understanding of the interplay between environmental changes and extinction dynamics during a past climate crisis, presenting an outlook for extinction threats in the worst-case "Shared Socioeconomic Pathways (SSP5–8.5)" scenario.

## Impact statement

What are the biggest consequences of climate change for marine ecosystems? Is it deoxygenation, thermal stress, ocean acidification or any combination thereof? The Permian–Triassic climate crisis was an episode of severe and rapid climate warming with similarities to the worst-case projected scenarios for the near future. To better understand which consequences of this climate event led to one of the most severe biodiversity crisis ever, we implemented a novel approach of statistically integrating high-resolution fossil data with high-resolution geochemical data. Our results demonstrate that for equatorial, marine ecosystems, oxygen isotope (temperature proxy) and cadmium isotope (primary productivity proxy) dynamics best explain the marine extinction. This suggests that the biggest threats to past and modern biodiversity in these settings are the impacts of thermal and nutrient stress, as well as associated trophic knock-on effects.

## Introduction

The most distinct and widely acknowledged causes of extinction in marine ecosystems today are pollution, habitat loss, overexploitation, introduction of invasive species and climate change (Bonebrake et al., 2019; IPBES, 2019). Understanding how these threats will reduce populations or drive species to extinction is a core component of modern-day conservation and policymaking. One issue is that horizon threats, like climate change, occur on global and centennial scales that are much broader in scope than knowledge that can be acquired using modern-day datasets alone (Bonebrake et al., 2019). The rock record, however, provides the only record of long-term biotic responses from disturbances and information on ecosystem re-establishment, which is now a priority in the Intergovernmental Panel on Climate Change (IPCC, 2021; Finnegan et al., 2023; Kiessling et al., 2023). We can, therefore, use different hyperthermal events of the past to provide key information on how horizon threats operate at community, ecosystem, and even biome levels.

The Permian–Triassic climate crisis is an exceptionally rapid warming event (around 8–12 °C rise in 60 ± 48 ka at low latitudes) from the latest Permian into the Early Triassic (late Griesbachian) (Joachimski et al., 2012, 2020; Sun et al., 2012; Chen et al., 2016; Gliwa et al., 2022). This climate crisis is thought to have been caused by the simultaneous eruptions of the Siberian Traps Large Igneous Province and the combustion of organic-rich sedimentary rocks

(Burgess and Bowring, 2015), leading to a large and rapid injection of $CO_2$ and volatiles into the atmosphere (Svensen et al., 2009; Joachimski et al., 2022). This event is also associated with the Permian–Triassic mass extinction, the most catastrophic mass extinction on Earth, which was highly selective against taxonomic groups that dominated pre-extinction marine communities (Foster et al., 2022a, 2023a), with an estimated loss of 81–96% of species (Erwin, 1993; Stanley, 2016). Multiple environmental perturbations occurred simultaneously during the climate crisis, making it difficult to disentangle which specific environmental changes were most significant in causing the extinctions. In addition, environmental stressors can interact in antagonistic or synergistic ways, where one stressor could reduce the impact of another or where a multiple of stressors can lead to an additive response (Benton, 2018; Penn et al., 2019). Furthermore, the drivers of extinction are expected to be spatially heterogeneous, as factors such as carbonate saturation state and the polar amplification of climate warming lead to heterogeneous patterns (Feldl and Merlis, 2021). Therefore, it is not unequivocally known exactly which factors played a major role in causing the biodiversity crisis. This lack of understanding is also due to the poor geographical coverage of continuous Permian–Triassic successions and the small number of sections that have been investigated at a high-resolution with multiple proxies for environmental and biodiversity changes.

Along the Meishan Hill, Zhejiang, China, a Permian–Triassic succession extends 2 km laterally and has been the subject of many paleontological and geochemical studies (Chen et al., 2015), which combined make the Meishan composite section the only place that can currently be quantitatively investigated to better understand which environmental proxies relate to biodiversity loss. In addition, Meishan's Permian–Triassic succession has a well-defined stratigraphic framework with each bed and sub-bed numbered allowing accurate correlations between studies performed over the last three decades. Furthermore, the Meishan D section has now been digitized into an open access, 3D interactive model that can be accessed via https://outcrop3d.deep-time.org/?model=194a20d7-958e-d871-11c2-ab18dfb62e16. In contrast, other regions with a rich paleontological and geochemical record for the Permian–Triassic transition, such as the Dolomites in Italy, do not yet have the same clear stratigraphic scheme or diversity in analyses that make studies like this one possible. During the Permian–Triassic transition, the Meishan section represents an outer slope setting in an equatorial (ca. 20°N) epicontinental sea (Yin et al., 2001). This means that the Meishan section can provide an analog into the causes of extinction during an extreme climate crisis for equatorial, shallow marine ecosystems (i.e., for the worst-case Shared Socioeconomic Pathways (SSP5–8.5) scenario, which predicts a total temperature increase of 3.3 to 5.7 °C by 2100 (IPCC, 2021). Here, we have conducted a data analysis by (a) creating a database of the fossil record to define the timing of extinction among different marine taxa, (b) assembled a database of 18 geochemical proxies (Figure 1 and Supplementary Table S1) for different environmental changes from the Meishan section that have been hypothesized to have had a critical role in the marine extinctions, and (c) quantitatively investigated which environmental changes associated with the climate crisis best explain the marine extinctions.

## Materials and methods

### Fossil data

Using the Geobiodiversity Database (http://www.geobiodiversity.com), Paleobiology Database (https://paleobiodb.org) and a

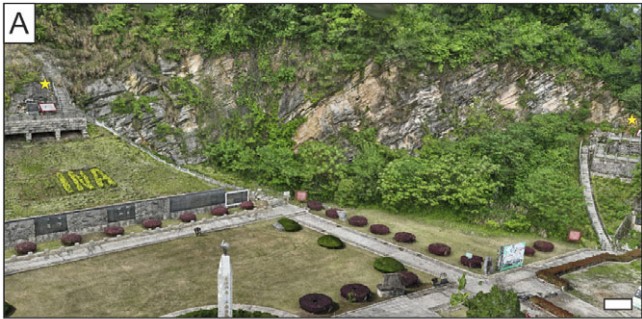

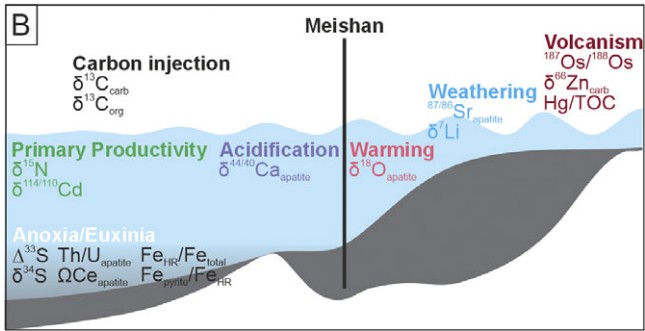

**Figure 1.** An outcrop and depositional model for the Meishan section. (A) The Deep-time Digital Earth 3D Outcrop model of the Meishan D section. The GSSP for the Wuchiapingian/Changhsingian and Permian/Triassic boundaries are indicated by stars. The interactive model can be accessed via https://outcrop3d.deep-time.org/?model=194a20d7-958e-d871-11c2-ab18dfb62e16. (B) Schematic of the paleoenvironmental setting, indicating the inorganic geochemical proxies that were selected to investigate the role of different environmental changes at Meishan. Data come from $\delta^7$Li (Sun et al., 2012), $\delta^{13}C_{carb}$ (Shen et al., 2013), $\delta^{13}C_{org}$ (Cao et al., 2009) supplemented at beds 23–38 with (Huang et al., 2007) and (Sial et al., 2021), $\delta^{15}$N (Cao et al., 2009), $\delta^{18}O_{apatite}$ (J. Chen et al., 2016), $\Delta^{33}$S and $\delta^{34}$S (Shen et al., 2011a), $\delta^{44/40}Ca_{apatite}$ (Hinojosa et al., 2012), $\delta^{66}Zn_{carb}$ (Liu et al., 2017), $^{87}Sr/^{86}Sr_{apatite}$ (Song et al., 2015), $Th/U_{apatite}$, $\Omega Ce_{apatite}$, (Song et al., 2012), $\delta^{114/110}$Cd (Zhang et al., 2018), $^{187}Os/^{188}Os$ (Liu et al., 2020), Hg/TOC (Sial et al., 2021), $Fe_{HR}/Fe_{tot}$ and $Fe_{py}/Fe_{HR}$ (Xiang et al., 2020). The bathymetry follows Zhang et al. (1997) at the time of the Permian/Triassic boundary.

literature search, we constructed a database of all known fossil occurrences from the Meishan section that spans from the Longtan Formation (Wuchiapingian) to the Nanlinghu Formation (Dienerian). The wide breadths of these time intervals were chosen to reduce the impact of edge effects. The clades included in the dataset were the Arthropoda, Brachiopoda, Bryozoa, Chlorophyta, Chordata, Cnidaria, Foraminifera, Mollusca, Radiolaria, Rhodophyta, and Problematica. The occurrences were manually vetted to ensure that typographic errors were corrected, so species did not appear with multiple spellings, and to ensure that individual species were not represented within multiple genera in the database due to taxonomic synonymy, in which case the most up-to-date species identification was followed. Freshwater and palynomorph fossils were removed.

To calculate the stratigraphic range of each species, occurrences of species with open nomenclature ("",?, aff., cf., informal) were taken into account. In older references from the Meishan section, the beds do not have the same stratigraphic subdivision as today and the occurrence of a specimen is considered present in all respective sub-beds. These occurrences that were not specified to a single bed/sub-bed as recognized in this study were subsequently excluded, as for them the timing of extinction is poorly constrained.

The resulting database included 603 species from 6,457 occurrences.

### Inorganic geochemical data

To investigate changes in environmental conditions, we downloaded the raw datasets of inorganic geochemical proxies for the Meishan section. We initially obtained all the articles for each proxy investigated for the Meishan section. Where multiple records of a single proxy were collected, we selected the most robust record, that is, we avoided mixing datasets collected from the same beds by different studies and selected the most extensive record. The resulting dataset included 18 proxies (Supplementary Table S1; Supplementary Figures S1–S4). The sample heights were standardized according to the Permian/Triassic boundary, with 0 cm marking the base of bed 27c. Data from the Meishan core, which is located 550 m west of the Meishan section D (GSSP section), were scaled to correlate with the section from Meishan D as the beds demonstrate considerable thickness variations.

### Statistical analysis

Determining the patterns of extinction can be confounded by subjective interpretations and by the Signor-Lipps effect. Therefore, to quantify the nature of extinction, here we used a modified version of the two-step extinction pulse algorithm of Wang and Zhong (2018) (see also extended materials and methods).

To quantitatively determine the number of breakpoints in the segmented regression analysis, the selgmented() function from the segmented package was used (Muggeo et al., 2014). The segmented () function was then used to statistically determine where these breakpoints occur for each geochemical proxy. For data imputation of the geochemical data, a segmented regression was used because (a) it is less affected by anomalous data points, (b) estimates are based upon overall trends in the data, (c) it does not assume that a relationship between different proxies exists, and (d) it recognizes significant shifts in data trends and is more dynamic than a single regression model (see also Supplementary Figures S5–S6).

We applied General Linear Models (GLM) with a Poisson distribution to test the effects of multiple geochemical proxies on changes of species richness through the study interval. Only proxies that showed significant correlations with species richness were included in the model, and model selection was carried out by exploring the value-inflated factors and factors that are highly correlated were successively dropped from the model. This resulted in two GLMs, one where $\delta^{114/110}Cd$ was dropped because it highly correlated with $\delta^{18}O_{apatite}$, and vice versa.

To investigate relationships between fossil incidence data and geochemical proxies, we carried out a partial-distance-based redundancy analysis using the Jaccard distance measure. Model selection was carried out using value-inflated factors, and factors that are highly correlated were successively dropped from the model. Variables that were insignificant in explaining incidence data dynamics using a permutation test for partial-dbrda were also dropped from the final model.

All analyses were carried in R v.3.4.3. Data and relevant code for this research work are stored and publicly available in GitHub: https://github.com/wjf433/QMEI

### Results

#### Nature of the mass extinction event

The nature of the Permian–Triassic mass extinction is hotly debated and has been interpreted either as a single pulse, interval,

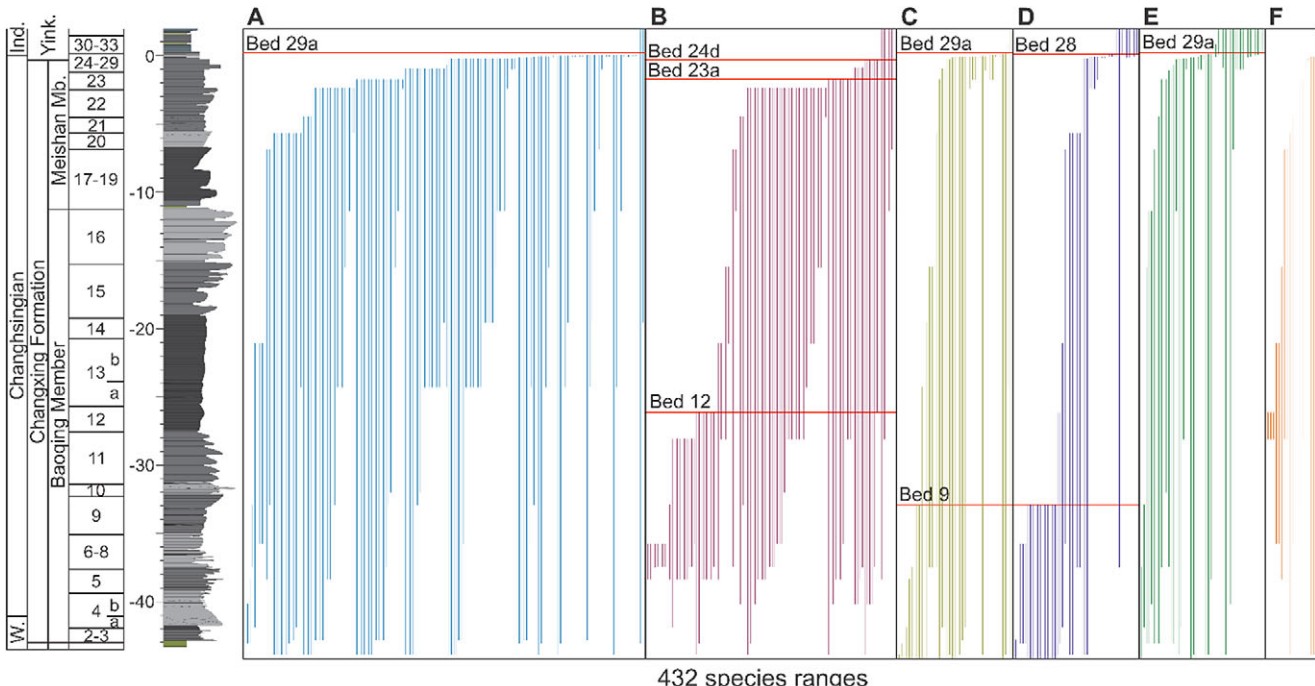

**Figure 2.** Stratigraphic ranges of fossil species (vertical lines) from the Meishan section. Stratigraphic ranges of (A) foraminifera, (B) arthropods (all ostracods, except one trilobite species), (C) brachiopods, (D) mollusks, (E) conodonts, and (F) other (includes: bryozoans, corals, calcareous algae, and *Tubiphytes*). Quantitatively determined extinction pulses for each phylum indicated (horizontal red line). Singletons are excluded from the figure and from determining the number of extinction pulses. Bed numbers and sedimentology follow Zhang et al. (1997) and Yin et al. (1995). 0 meters is taken as the base of bed 27c, which is the biostratigraphic position of the Permian/Triassic boundary that is defined by the first appearance of *Hindeodus parvus* (Yin et al., 2001). W. = Wuchiapingian, Ind. = Induan, Yink. = Yinkeng Formation.

or a two-pulsed extinction (Jin et al., 2000; Shen et al., 2011a; Song et al., 2014; Wang et al., 2014). Quantifying the number of pulses of extinction, which considers confidence intervals of stratigraphic ranges (Figure 2 and Supplementary Figures S7–S11), demonstrates that the nature of the mass extinction is complex and varies between different phyla. Near the Permian/Triassic boundary, the traditional extinction horizon (bed 25) (Wang et al., 2014) marks numerous last occurrences (LAD), leading to composition shifts (Figure 2 and Supplementary Figures S7–S11). Our analysis shows, however, that a single pulse of extinction (the final LADs) occurs at bed 28 for mollusks and bed 29a for foraminifera, brachiopods, and conodonts (Figure 2). Ostracods, instead, record two earlier pulses of extinction at beds 23a and 24d (Figure 2), the latter coinciding with a sequence boundary. The species richness of the remaining groups is too low to detect the timing of extinction, but the highest occurrences do not occur above bed 27c. This suggests that the nature of the mass extinction event at Meishan, except for ostracods, is best characterized as an extinction interval (51 cm), from beds 25 to 29a (*C. meishanensis* – *I. isarcica* conodont zones). Such an interpretation is also relatively consistent with a stark reduction in bioturbation and tiering depth at the base of bed 25 (Zhao and Tong, 2010).

On the contrary, ostracods record an earlier major extinction interval from beds 22-23a and a subsequent minor pulse at bed 24d (Crasquin et al., 2010), suggesting that this group of organisms was more sensitive to initial environmental changes or responded to different environmental changes before the main extinction interval. When the timing of extinction is investigated with all the species included, the mass extinction event is consistent between the different data splits, with a 2-pulse event at beds 23a and 29a, with bed 23a reflecting the selective extinction of ostracods (Supplementary Table S2). In addition to the extinction interval that spans the Permian/Triassic boundary, ostracods, mollusks, and brachiopods also record a minor extinction pulse earlier in the Changhsingian (beds 9 and 12, Figure 2).

These changes are also reflected by the breakpoints and a rapid decline in species richness at beds 23, 25, and 29a (Figure 3). Lithology changes and sequence stratigraphic boundaries play a role in determining the stratigraphic position of the LAD (Holland and Patzkowsky, 2015; Nawrot et al., 2018; Zimmt et al., 2021). At Meishan, the extinction interval includes a lithostratigraphic boundary at the base of bed 25, and a transgressive surface at bed 27a, suggesting that these sedimentological changes affect our interpretations of the nature and timing of the extinction. Despite that, radiometric dating proposes that beds 25 to 28 only represent 60 ± 48 ka (Burgess et al., 2014), and any hiatuses associated with sequence stratigraphic surfaces during the extinction interval are of relatively short duration.

Changes in species richness, in particular the earlier onset of ostracod extinctions, are problematic when trying to compare extinctions with geochemical proxies. This is because many of the proxies that have been investigated at the Meishan section only span a short interval, for example, $\delta^{114/110}$Cd only spans beds 22–33 (Zhang et al., 2018), after species diversity has already started to decline (Figure 3). Analyses linking geochemical and fossil data were, therefore, restricted to beds 22–29a. Investigations did not extend beyond bed 29a because the protracted low diversity after the extinction interval can be attributed to a delayed recovery rather than environmental conditions.

## Quantifying the causes of extinction

Quantifying the causes of extinction is complex, as environmental changes will manifest with different patterns and may be reflected

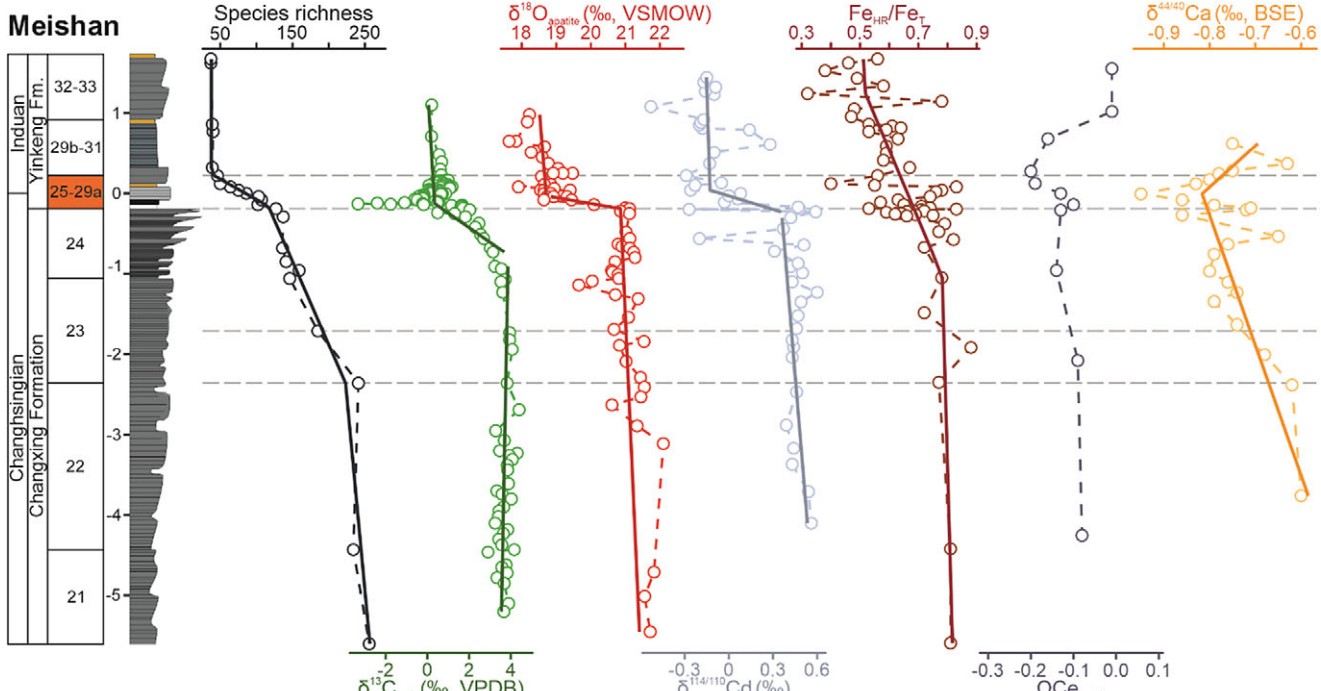

**Figure 3.** Stratigraphic correlation of selected paleoenvironmental proxies with species diversity at the Meishan section, South China, with segmented regression lines overlain. $\delta^{13}$C$_{carb}$ (Shen et al., 2013), $\delta^{18}$O$_{apatite}$ (VSMOW) (Chen et al., 2016), $\delta^{114/110}$Cd (Zhang et al., 2018), Fe$_{HR}$/Fe$_{tot}$ (Xiang et al., 2020), $\Omega$Ce$_{apatite}$ (Song et al., 2012), $\delta^{44/40}$Ca$_{apatite}$ (Hinojosa et al., 2012). The main extinction interval (beds 25–29a) is highlighted in orange and with two horizontal dashed lines. Note: only paleoenvironmental proxies that showed significant relationships with diversity are included, for a full figure with all the paleoenvironmental proxies see Supplementary Figures S1–S4.

as either state-shifts, anomalies, or correlations that can be associated with biodiversity dynamics. For example, the investigated proxies associated with volcanism, for example, Hg/TOC, $\delta^{66}$Zn and $^{187}$Os/$^{188}$Os, are expected to appear as anomalies or spikes. The proxies Hg/TOC, $\delta^{66}$Zn and $^{187}$Os/$^{188}$Os show anomalies that coincide with the onset of the mass extinction interval (Supplementary Figure S3), with the Hg/TOC, $\delta^{66}$Zn, and $^{187}$Os/$^{188}$Os anomalies from beds 24b-24e being interpreted to reflect volcanism associated with the Siberian Traps coming along with input of volcanic ashes (Liu et al., 2017; Liu et al., 2020; Sial et al., 2021).

A segmented regression analysis, which can be used to quantify significant temporal shifts in proxies (i.e., state-shifts), recognizes significant changes for $\delta^{13}$C$_{carb}$, $\delta^{18}$O$_{apatite}$, $\delta^{114/110}$Cd, and $\delta^{15}$N at the onset of the extinction interval (bed 25, Figure 3 and Supplementary Figures S1–S4). In addition, TOC shows a breakpoint at bed 22 (Supplementary Figure S1), corresponding with the main extinction pulse of ostracods (Figure 3). However, Th/U$_{apatite}$ ratios show a state shift at bed 29 (Supplementary Figure S4), with an interpreted state shift from oxic to anoxic conditions (Song et al., 2012), and corresponding with a plateau of low richness.

One issue with comparing the different proxies and changes in species richness or incidence data is the difference in resolution between the different datasets. To allow for statistical exploration of the data, the data were aggregated to the same resolution as the species dataset, that is, bed and sub-bed level resolution. In addition, not all the beds record proxy values, and, therefore, data were interpolated using the segmented regression curves for each proxy (Supplementary Figures S5 and S6). Another issue is that many of the different geochemical proxies correlate with one another (Supplementary Figures S12–S14), and these correlations are not necessarily direct causal effects, but could rather be associated with autocorrelation effects within each time series, indirect links, or common drivers (Runge et al., 2019). This makes it difficult to disentangle whether the proxy is robust enough to interpret environmental changes, if an environmental change is causing a decline in diversity, or both diversity and proxy dynamics have a common cause. A correlation plot shows that $\delta^{13}$C$_{carb}$, $\delta^{18}$O$_{apatite}$, $\delta^{114/110}$Cd, and $\delta^{15}$N are significantly correlated, which suggests these proxies share a common cause.

Many of the proxies show a correlation with changes in species richness (Supplementary Table S3). A Poisson regression model was performed to identify which proxies best explain the diversity

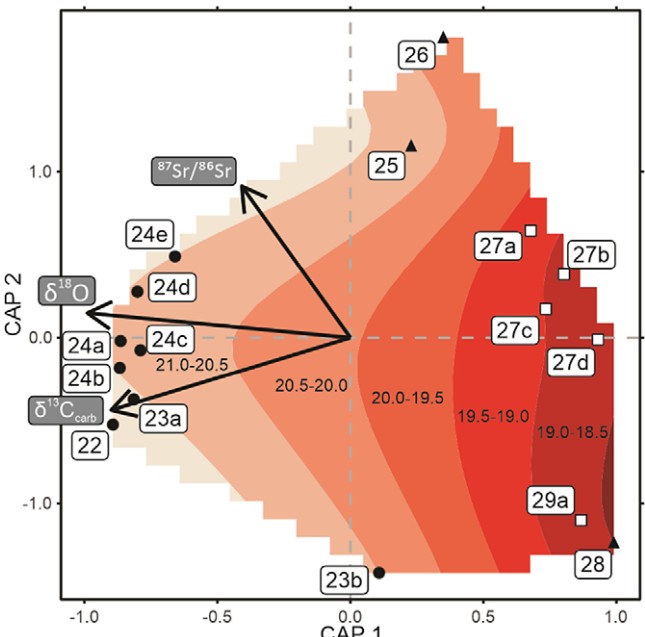

**Figure 4.** Partial-Distance-based Redundancy Analysis (capscale) for fossil assemblages and geochemical proxies from the Meishan section. Included vectors are the geochemical proxies that were determined as having a significant relationship with the fossil assemblages. Sample point shapes relate to bed lithology: filled circles = limestone, open squares = silty limestone, and filled triangles = clay. The bed numbers for each assemblage are indicated, and only beds 22-29a are included due to limited coverage of geochemical proxies at the Meishan section. Smooth contours of the oxygen isotope values underlie the ordination plot to demonstrate the relationship with the fossil assemblages.

dynamics. Value-inflated factors show that the correlation between $\delta^{13}$C$_{carb}$, $\delta^{18}$O$_{apatite}$, $\delta^{114/110}$Cd, and $\delta^{15}$N significantly affects the quality of the model. For this reason and because the $\delta^{15}$N data is at a low resolution, $\delta^{15}$N was dropped from the model, whereas for $\delta^{114/110}$Cd and $\delta^{18}$O$_{apatite}$ two separate models were run. The generalized linear models show that $\delta^{13}$C$_{carb}$, $\delta^{18}$O$_{apatite}$, $\delta^{114/110}$Cd, and $\delta^{44/40}$Ca$_{apatite}$ have significant relationships with changes in species diversity at Meishan (Table 1). In addition, no proxies showed a significant relationship between proxy variance and extinction rate.

A partial-distance-based redundancy analysis (partial-dRDA) was undertaken to investigate the changes in fossil incidence data

**Table 1.** Generalized linear model of significant environmental variables (geochemical proxies) and changes in diversity

| Model | Parameter | Estimate | 95% Confidence intervals | | t-value | p-value |
|---|---|---|---|---|---|---|
| | | | 2.5% | 97.5% | | |
| Species diversity pseudo–$R^2$ = 0.62 | (Intercept) | 0.40 | −1.35 | 2.15 | 0.44 | 0.658 |
| | $\delta^{13}$C$_{carb}$ | 0.06 | 0.01 | 0.11 | 2.49 | **0.001** |
| | $\delta^{18}$O$_{apatite}$ | 0.25 | 0.17 | 0.32 | 6.38 | **< 0.001** |
| | $\delta^{44/40}$Ca$_{apatite}$ | 0.94 | 0.22 | 1.67 | 2.55 | 0.011 |
| Species diversity pseudo–$R^2$ = 0.59 | (Intercept) | 5.71 | 5.13 | 6.30 | 19.18 | **< 0.001** |
| | $\delta^{13}$C$_{carb}$ | 0.03 | −0.03 | 0.09 | 1.08 | 0.279 |
| | $\delta^{114/110}$Cd | 0.92 | 0.57 | 1.28 | 5.09 | **< 0.001** |
| | $\delta^{44/40}$Ca$_{apatite}$ | 1.61 | 0.92 | 2.31 | 4.55 | **< 0.001** |

*Note*: Model selection was based on proxies that showed consistent and significant linear relationships with diversity (Supplementary Table S3). $\delta^{114/110}$Cd and $\delta^{15}$N were dropped from the first model because they showed a significant correlation with $\delta^{18}$O$^{apatite}$ that negatively impacted the model (Supplementary Material). A second model swapping $\delta^{18}$O$^{apatite}$ and $\delta^{114/110}$Cd was done to investigate the best model.

for beds 22 to 29a and changes in geochemical proxies (Figure 4). Once more, value-inflated factors show that the correlation between $\delta^{13}C_{carb}$, $\delta^{18}O_{apatite}$, $\delta^{114/110}Cd$, and $\delta^{15}N$ significantly affects the quality of the model. $\delta^{114/110}Cd$ and $\delta^{15}N$ were, therefore, dropped from the model. The partial-dRDA showed that $\delta^{13}C_{carb}$, $\delta^{18}O_{apatite}$, and $^{87}Sr/^{86}Sr$ best-explained changes in the incidence data. Swapping $\delta^{114/110}Cd$ with $\delta^{18}O_{apatite}$ shows that $\delta^{18}O_{apatite}$ is a more significant proxy for explaining incidence data dynamics. When only the significant factors are included in the partial-dRDA model, only $\delta^{13}C_{carb}$ and $\delta^{18}O_{apatite}$ record significant relationships (Figure 4). It is also evident that the fossil incidence data cluster according to lithology (Figure 4), highlighting how lithological changes reflect changes in the environment affecting species loss.

## Discussion

Due to the large suite of geochemical proxies investigated for the Meishan section, a number of different environmental changes have been proposed as possible causes of the Permian–Triassic mass extinction. Our quantitative analysis, combining diversity and proxy data, demonstrates that $\delta^{13}C_{carb}$, $\delta^{18}O_{apatite}$, and $\delta^{114/110}Cd$ are key for understanding the cause(s) of the mass extinction event. The geochemical signature of these proxies is mostly generated in the euphotic zone with a high chance of transfer into sediments without much alteration, suggesting that the relationships between these proxies and the fossil record reflect past environmental-life interactions. In addition, the lack of relationship between the fossil incidence data and the geochemical proxies that are affected by lithology changes supports that our interpretations are robust. $\delta^{13}C_{carb}$ is often interpreted as reflecting the release of large quantities of isotopically light carbon into the atmosphere, changes in primary productivity, and changes in carbon burial rates (Cao et al., 2002). $\delta^{13}C_{carb}$ can, therefore, signify an environmental disturbance and even the trigger of the mass extinction (Cui et al., 2015), but it cannot be inferred to identify the underlying environmental changes that drove species to extinction. In addition, $\delta^{13}C_{carb}$, $\delta^{18}O_{apatite}$, and $\delta^{114/110}Cd$ are significantly correlated, which we infer as being impacted by a common cause.

A negative excursion in $\delta^{18}O_{apatite}$ is interpreted to reflect a rapid, 8–12 °C, warming associated with the mass extinction event at Meishan (Joachimski et al., 2012; Sun et al., 2012; Chen et al., 2016) and is consistently the best explanatory factor for diversity dynamics at Meishan (Table 1 and Figure 4). Thermal stress is understood to limit the performance of aerobic marine organisms, because the pejus temperature is close to the temperature optimum on the upper thermal limit, and increasing temperatures beyond a marine organism's optimum temperature range rapidly lead to a reduction in the aerobic scope of marine organisms (excess capacity supporting activity, growth, and reproduction) (Pörtner, 2012; Pörtner et al., 2017). An expectation from this mechanism would be an observed decrease in body size as temperatures increase and primary productivity declines. The only body size data from Meishan with enough measurements comes from two species of foraminifera, *Diplosphaerina inaequalis* and *Frondina permica* (Song et al., 2011), that record decreasing size with more negative $\delta^{18}O_{apatite}$ values (Figure 5), suggesting a decrease in aerobic scope before their LADs in beds 27c and 29a, respectively. The paleoequatorial setting of the Meishan section also means that this locality would also have experienced some of the highest climate velocities (sensu Burrows et al., 2011) with Earth system models for the Permian–Triassic mass extinction demonstrating the consequent

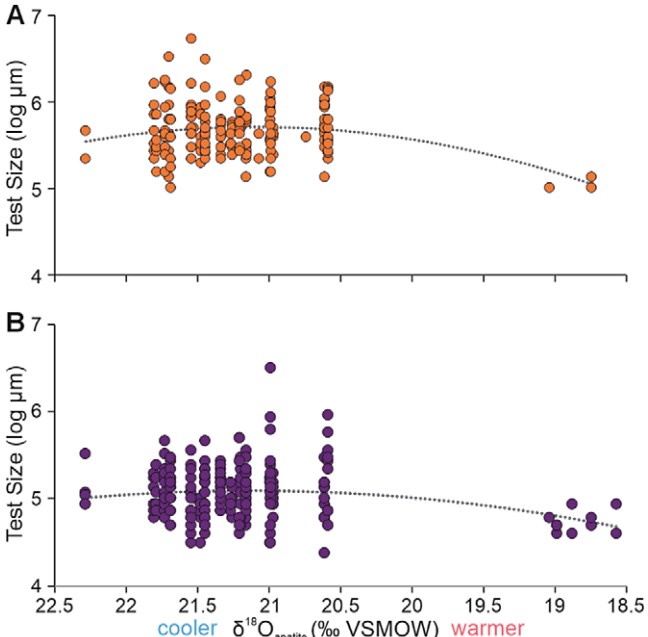

**Figure 5.** A scatter plot showing the relationship between foraminifera test size and $\delta^{18}O_{apatite}$ (a temperature proxy) from the Meishan section. (A) Measurements of *Frondina permica* from beds 13a-27c (B) Measurements of *Diplosphaerina inaequalis* from beds 13a-29a. An order 2 polynomial trend line is underlain to illustrate the relationship between $\delta^{18}O_{apatite}$ values and test size. As $\delta^{18}O_{apatite}$ values decrease they can be used to infer a warming of the climate and vice versa, which is indicated on the axis. The test sizes of the foraminifera were converted to geometric sizes and log-transformed. Body size data are from Song et al. (2011) and oxygen isotope data from Chen et al. (2016). Other species could not be included due to either the lack of species-level identification or size measurements.

loss of aerobic habitats able to support the metabolism of marine ectotherms (Penn et al., 2019). This idea is supported, by the poleward migrations of Permian holdover radiolarians, sponges, and conodonts away from equatorial settings associated with the warming (Foster et al., 2023b). The significant relationship between $\delta^{18}O_{apatite}$ with diversity, compositional changes, body size and poleward migrations supports the interpretation that temperature-driven hypoxia was fundamental in causing equatorial extinctions/extirpations during the Permian–Triassic climate crisis.

$\delta^{18}O_{apatite}$ is also significantly correlated with $\delta^{114/110}Cd$ and $\delta^{15}N$, where both $\delta^{114/110}Cd$ and $\delta^{15}N$ record negative excursions that are interpreted to reflect primary productivity dynamics (Cao et al., 2009; Zhang et al., 2018). $\delta^{114/110}Cd$ reflects nutrient utilization by phytoplankton and is an indirect proxy for primary productivity. At Meishan a negative excursion in $\delta^{114/110}Cd$ coincides with the mass extinction interval reflecting a collapse in primary productivity (Figure 2; Zhang et al., 2018), which is also associated with the major extinction of radiolarians (the primary fossil record of planktic biodiversity) (O'Dogherty et al., 2010). A stark reduction in primary productivity would be catastrophic for marine ecosystems because the cascading effect of extinction causes knock-on effects on species populations in successive layers of the marine food web (Huang et al., 2023). Huang et al. (2023) inferred that the loss of radiolarians at the onset of the extinction interval could have been a cascading effect of a collapse in primary productivity. The migration of radiolarians to thermal refugia at higher latitudes and deeper waters (Foster et al., 2023b) could also suggest additive effects of different environmental stressors, that is, nutrient and thermal stress. In addition, aerobic metabolism is not only affected

by thermal stress but also nutritional stress, which can exacerbate the effects of climate change in marine ectotherms (Saulsbury et al., 2019). The impacts on the aerobic metabolism of marine organisms can also be inferred from the observed decrease in the body size of surviving taxa (He et al., 2010).

The hypothesis that primary productivity collapsed during the Permian–Triassic mass extinction is controversial, where some proxies suggest a collapse (e.g., Zhang et al., 2018) while others suggest enhanced primary productivity (e.g., Qiu et al., 2019). In part, this can be explained by spatial heterogeneity in primary productivity rates as evidenced by spatial variations in productivity proxies (e.g., Shen et al., 2015), but conflicting results are also known from the Meishan section (e.g., Zhang et al., 2018; Qiu et al., 2019). $\delta^{114/110}Cd$ is not a redox proxy, but in sulfide-bearing anoxic sediments there are more negative values than in oxic surface waters (Hohl et al., 2017). Zhang et al. (2018), however, noted that there is no correlation between sulfur concentrations and $\delta^{114/10}Cd$ values, inferring that the inferred trends reflect changes in primary productivity. Other proxies for primary productivity come from lipid biomarkers and $\delta^{15}N$, which show changes in the archaeal and bacterial communities as a consequence of the environmental changes (Xie et al., 2005; Cao et al., 2009). Taken together, this suggests that not only did primary productivity appear to collapse in South China, but there were also phytoplankton community shifts which would have led to nutrient stress.

In the equatorial paleosetting of the Meishan section, both thermal and nutrient stress are interpreted to best explain the extinctions. It has been shown that despite inter-specific differences, there are clear differences in hypoxia tolerance among higher taxa (Song et al., 2013). Ostracods and crustaceans have the least tolerance to hypoxia compared to other invertebrate groups (Song et al., 2013), and their earlier onset of extinction at the Meishan section also corresponds to the initial changes in the $\delta^{18}O_{apatite}$ negative excursion (Figure 2). $\delta^{18}O_{apatite}$ dynamics have been divided into two phases associated with different rates and magnitudes of warming (Wu et al., 2023), with the first phase, coinciding with the major extinction of ostracods, being slower and of a smaller magnitude. Pre-extinction changes in $\delta^{18}O_{apatite}$ in equatorial settings have also been related to body size changes in ammonoids (Gliwa et al., 2022) and correspond to pre-extinction changes in brachiopod assemblages (Zhang et al., 2017). This suggests that pre-extinction slower warming and the following rapid warming led to different timings of extinction for different marine organisms, depending on their sensitivity to temperature and oxygen-concentration changes.

Widespread anoxic conditions throughout shallow and deep marine basins have long been associated with marine extinctions. The Meishan section has been the subject of several paleoredox studies, utilizing: pyrite framboids (Wei et al., 2020; Chen et al., 2015), sulfur isotopes (Shen et al., 2011b) as well as iron speciation, and redox-sensitive metals (Xiang et al., 2020). Iron speciation has been established as a proxy for local water column redox in clastic successions and some carbonate successions, yet the $Fe_{HR}/Fe_{tot}$ data for Meishan records almost persistently anoxic conditions for bed 21 to 34 (Xiang et al., 2020). Considering the abundant and deeply penetrated trace fossils (>20 cm depth) found throughout most of this interval (Zhao and Tong, 2010), the interpretation of consistently anoxic conditions is equivocal. Hence, the anoxic $Fe_{HR}/Fe_{tot}$ signals at best represent dynamic redox conditions with intermittent anoxic intervals or an alternative explanation is the redeposition of dissolved Fe at Meishan that was released from nearby oxygen minimum zones (similar to the Guaymas Basin in the Gulf of California; Scholz et al., 2019). Multiple sulfur isotope signals

from pyrite and pyrite aggregate sizes are suggested to support the development of episodic anoxic water column conditions during the deposition of beds 22–24 (Shen et al., 2011b). Depending on the extent and duration of these anoxic episodes, they could have contributed to the loss of ostracods before the main extinction interval, as deposit-feeding ostracods are negatively impacted by falling oxygen levels (Lethiers and Whatley, 1994). Conversely, $\Omega Ce_{apatite}$ anomalies from the same beds mean deposition was in an oxygenated setting (Song et al., 2012). Above bed 24d, redox-sensitive metal enrichment factors record decreasing trends (Xiang et al., 2020), reflecting increasing oxygenation of the water column. This is also supported by pyrite $\delta^{34}S$ and $\Delta^{33}S$ signals that do not support an anoxic water column interpretation (Shen et al., 2011b). Overall, the water column redox conditions at Meishan were likely dynamic during the Permian–Triassic transition, but the timing and extent of anoxic episodes are too poorly constrained to unequivocally conclude that anoxia or dysoxia played a role in the relatively shallow setting of Meishan.

Despite the intense geochemical and paleontological research on the Meishan section, this study highlights some limitations that must be addressed in future research. The restriction of investigations of geochemical proxies over a short interval at the Permian/Triassic boundary hinders our ability to understand how environmental conditions evolved over the Changhsingian and how that relates to the climate crisis (e.g., $\delta^{44/40}Ca_{apatite}$; Hinojosa et al., 2012). Therefore, even though $\delta^{44/40}Ca_{apatite}$, a potential proxy for ocean pH (Hinojosa et al., 2012), is recorded as having a significant relationship with changes in species richness (Table 1), the short record still makes this interpretation equivocal. Even if the $\delta^{44/40}Ca_{apatite}$ trends are seen as robust, concerns of using this proxy to determine ocean acidification have been raised (Komar and Zeebe, 2016; Foster et al., 2022b), and, therefore, the role of ocean acidification in the extinctions for these settings is still unknown. In addition, the lack of abundance data and other ecological data from paleontological studies (e.g., Song et al., 2009; Crasquin et al., 2010) means it is not yet possible to investigate the ecological impacts of the Permian–Triassic climate crisis beyond the timing of extinction. Therefore, several ecological changes, such as changes in relative abundance, dominance, or body size and how they relate to environmental changes, cannot yet be explored. Finally, the cause(s) of extinction is expected to vary on various spatial scales, and more high-resolution studies from other sections and regions are, therefore, required. Despite these shortcomings, our statistical analysis demonstrates that the extreme impact of environmental changes on the aerobic metabolism of marine ectotherms and the cascading effects of extinction best explain the cause of extinction in epicontinental, equatorial settings. This means that for the worst-case RCP scenario, the biggest climate threats to modern-day shallow marine, equatorial biodiversity are thermal and nutrient stress.

**Open peer review.** To view the open peer review materials for this article, please visit http://doi.org/10.1017/ext.2024.9.

**Supplementary material.** The supplementary material for this article can be found at http://doi.org/10.1017/ext.2024.9.

**Data availability statement.** All analyses were carried in R v.3.4.3. Data and relevant code for this research work are stored and publicly available in GitHub: https://github.com/wjf433/QMEI.

**Acknowledgements.** We would like to thank Steve Wang (Swathmore College) for his work and help in developing the parallelization of his extinction pulse algorithm. We would like to thank the CEN-IT at Universität Hamburg for access to the computer cluster required for our analyses. We would like to thank

all the scientists who have entered fossil occurrence data into both the Geobio-diversity Database and Paleobiology Database. This is Paleobiology Database official publication No. 487 and Geobiodiversity Database official publication No. 2. W.J.F. and X.W would like to thank Prof. S-Z. Shen and Dr. H. Xu (Nanjing University) for taking and introducing them to Meishan. We would like to thank Prof. H. Song for providing the foraminifera body size data. We would also like to acknowledge the entire Deep-time Digital Earth Outcrop3D team for digitizing the Meishan section.

**Author contribution.** W.J.F. conceived the project. W.J.F., A.B.F., and Q.L. collected the data. W.J.F. wrote the code and ran the analysis. W.J.F. wrote the initial manuscript draft and all authors contributed thereafter.

**Financial support.** This research was funded by the DFG Grant FO 1297/1 awarded to W.J.F.

**Competing interest.** The authors declare no competing interests exist.

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
