## [Editor Report]

I agree with the reviewers that the manuscript is well developed and with a good potential to become an important contribution to the field, but I also agree that there is room for improvement.

Reviewer 1:

This is a nice review, integrating the best available data on the ‘type’ marine section through the Permian-Triassic boundary, at Meishan. The boundary geology and environmental proxies have been described in detail before (e.g. Chen et al. 2015), but the new analysis uses statistical methods to estimate the actual shape of the geochemical proxy time series – the segmented method, rarely used in this context, is a smart way of identifying sharp changes in phenomena a posteriori – it nicely shows the truly sharp shifts at the PTB. Also, the linear model fits between biotic and abiotic signals are original and much better than the usual kinds of wualitative wiggle matching methods.

In quoting temperature increases at the PTB of 8–10 oC, Sun et al. (2012) is cited; is this the most recent reliable reference? Surely, Joachimski et al. (2022) ought to be mentioned, as well as any other such recent references, and with a higher range of temperature rises, up to 15 oC? If these higher temperatures, and Joachimski and other recent papers are unreliable, you need to say so, and why.

Likewise, some of the other cited sources are quite old – like Pörtner et al. (2012) and Roopnarine et al. (2006), where both authors have published much more recent and in-depth studies of responses of marine animals to heat stress and ecosystem modelling across the PTB respectively. In particular, Huang et al. (2023; Current Biology 33, 1059-1070. e4 (doi: 10.1016/j.cub.2023.02.007) ) explore ecosystem stability in South China marine sections through the PTB interval, extraordinarily relevant to this paper. Further, Benton (2018, Phil. Trans. R. Soc.) reviews the literature of stressors on marine life, comparing modern physiology studies (such as Pörtner) with PTB environmental stressors.

Maybe do a quick Google-Scholar search of post-2015 or post-2020 literature to make sure the latest papers are always cited.

Quite a few careless typos:

2/5: Is it, = Is it

2/13 and 16: ‘that’ twice – only needed once

2/42: define RCP, and preferably avoid acronym in Abstract

4/13: explain RCP on first usage

4/16: (603 species from 6457 occurrences) =- delete here, as properly explained at 5/36 [ delete one or other anyway to avoid repetition]

5/19: vetted corrected = corrected

5/22: species identification of the species = identification of the species

5/32; 10/33: subbed – meaning? explain… ?subsetted?

5/48: samples = sample

6/11: selgmented() = segmented()

7/21; 10/20: Whereas = On the other hand [or you can use ‘whereas’ within the same sentence to give an opposite case]

9/48: because of = because the

10/35–37: if = whether [3 times]

12/36: Smooth contour… underlay… = Smooth contours… underlie…

13/10–11: which we infer as being impacted by a common cause = which we take as evidence for a common cause – or some such. But be clear – are you saying one of these signals controls the others, or that there is an unknown additional driver somewhere out there that drives all three?

13/24: Portner = Pörtner

15/28: short-interval = short interval

Reviewer 2:

This paper systematically summarizes the biological changes and paleoenvironmental changes at the Permian-Triassic boundary interval in the GSSP Meishan section. Based on some quantitative methods, the causes of this mass extinction are explored, and it is a better summary paper. I recommend publication after moderate revisions.

Major issues:

This paper has some limitations in discussing the causes of Permian-Triassic extinction using only one section. Changes in conditions like redox conditions and productivity of the ocean are characterized by regional changes, and it is difficult to represent global changes and discuss the real reason of mass extinction. Because mass extinction is a global phenomenon.

The changes of δ114Cd data is not only caused by productivity, but some other causes. For example, the redox conditions of the ocean, can also cause changes in δ114Cd, so caution should be preserved in interpreting the effect of productivity on the Permian-Triassic extinctions. In addition, several studies have shown that changes in marine productivity during the Permian-Triassic are highly controversial, with both the view that productivity increased and the view that productivity decreased (e.g. Twitchett 2001; 2007; He Weihong et al 2015 vs Qiu Zhipu et al 2019 and references therein) .

Minor issues:

Citations are not standardized, e.g. Chen ZQ vs Chen. I know these are two different authors, but standardize the format.

Meishan section or sections. In the main text the authors all use the latter, which is clearly wrong.

“18 geochemical proxies”, but in Figure 1, I only find 17 proxies.

Page 6 lines 9-11, “selgmented()”? Please give some explanation

The source of the fossil data in Figure 2 needs to be given.

Pages 14 Lines 24-25 a mistake: δ\δ18Oapatite

The productivity indicators given in the paper include δ114Cd and δ15N, why only isotopic indicators were chosen and not elemental ones, and also, between these two indicators, why only δ114Cd responded strongly with species abundance but not δ15N, can it be directly attributed to extinction due to nutrient utilization pressure? At the same time, it is debated whether primary productivity declined at the end of the Permian.

References:

Page 16 line 44, PNAS should be used in full name

Page 20 line 18, 2013 not 2014

Page 22 line 10, Episodes Journal of International Geoscience or Episodes

---

## [Editor Report]

Thank you for submitting the revised version of your manuscript. Both reviewers are satisfied with the revision done, and based on the evaluation of the changes made in the manuscript and comments of reviewers, I think that the manuscript can be accepted following one minor edit suggested by one of the reviewers:

Some figures are referenced out of order in the texts, e.g. the first cited figures is Figs S9-10.

---

## [Editor Report]

Thank you for submitting the revised version of your manuscript. Based on the evaluation of the edits made in the manuscript I think that the manuscript can be accepted in its present form.

---

## [Editor Report]

Thank you for submitting the revised version of your manuscript. Based on the evaluation of the changes made in the manuscript I think that the manuscript can be accepted in its present form.